# Inherent Consistent Learning for Accurate Semi-supervised Medical Image Segmentation

**Ye Zhu**[1]                                                                    ZHUYE1@CUHK.EDU.CN
**Jie Yang**[1]                                                              JIEYANG5@LINK.CUHK.EDU.CN
**Si-Qi Liu**[1]                                                                      SIQILIU@SRIBD.CN
**Ruimao Zhang**[1,*]                                                       RUIMAO.ZHANG@IEEE.ORG

[1] *Shenzhen Research Institute of Big Data, The Chinese University of Hong Kong(Shenzhen), China*
[*] *Corresponding author*

**Editors:** Accepted for publication at MIDL 2023

## Abstract

Semi-supervised medical image segmentation has attracted much attention in recent years because of the high cost of medical image annotations. In this paper, we propose a novel Inherent Consistent Learning (ICL) method, aims to learn robust semantic category representations through the semantic consistency guidance of labeled and unlabeled data to help segmentation. In practice, we introduce two external modules, namely Supervised Semantic Proxy Adaptor (SSPA) and Unsupervised Semantic Consistent Learner (USCL) that is based on the attention mechanism to align the semantic category representations of labeled and unlabeled data, as well as update the global semantic representations over the entire training set. The proposed ICL is a plug-and-play scheme for various network architectures, and the two modules are not involved in the testing stage. Experimental results on three public benchmarks show that the proposed method can outperform the state-of-the-art, especially when the number of annotated data is extremely limited. Code is available at: https://github.com/zhuye98/ICL.git

**Keywords:** Semi-supervised Learning, Medical Image Analysis, Semantic Segmentation.

## 1. Introduction

Anatomical organ or tumor segmentation has aroused extensive attention in recent years due to the helpful pixel/voxel-wise visual guidance in various medical imaging analysis tasks like disease diagnosis and radiation therapy (Wang et al., 2018; Sahiner et al., 2019). With the evolution of deep learning techniques, popular networks have also been applied in this field and achieved huge success (Strudel et al., 2021; Chen et al., 2021a; Hatamizadeh et al., 2022). However, these approaches are highly dependent on large-scale annotated data due to the data-driven nature of deep networks. Different from natural images, the annotation costs of medical images are much higher since it must be done by well-trained doctors (Jiao et al., 2022; Ji et al., 2022).

To mitigate the need of a large amount of annotated data, Semi-Supervised Learning (SSL) has been introduced to learn from scarce annotated data and become a crucial and challenging problem in the medical imaging analysis community. Existing commonly used semi-supervised medical image segmentation methods are mainly related to pseudo-labeling (self-training), co-training, and consistency regularization. The self-training strategy that relies on generated pseudo labels as data expansion was first proposed by (Blum and

Mitchell, 1998) and later followed by other variants (Zou et al., 2020; Chen et al., 2021b); Co-training methods (Blum and Mitchell, 1998; Zhou et al., 2019; Peng et al., 2020; Wang et al., 2021; Luo et al., 2022a) assume that different models can learn multiple independent and complementary views from the same data. By combining information from different perspectives, the accuracy of the final classification can be greatly improved. Consistency regularization on intermediate representations of input perturbation was also introduced to semi-supervised learning (Bachman et al., 2014; Ouali et al., 2020; Luo et al., 2022b) to robustness. However, most follow the standard solution that relies on low-level features for consistency regularization or co-training while ignoring the global high-level features and their rich semantic information. For instance, (Luo et al., 2022a) leverages different learning paradigms of CNN and Transfomer to perform consistency regularization on the output-level pseudo labels. Although the method achieved the current SOTA performance, experimental results show its limitation in discriminating some specific categories, especially when the background appears similar to the foreground categories.

To tackle the above issues by using the limited annotated training samples, this paper proposes a novel semantic consistent learning scheme termed Inherent Consistent Learning (ICL), which can more effectively leverage unlabeled data to assist in learning robust category representations. In practice, we introduce two external modules, Supervised Semantic Proxy Adaptor (SSPA) and Unsupervised Semantic Consistent Learner (USCL), aligning the semantic representations of labeled and unlabeled data and updating the global category representations. In practice, the global category representations (*i.e.*, semantic-aware proxies) are initialized as the learnable parameters of SSPA and updated at multiple scales by using the feature maps of the labeled data. These updated category representations are further used to interact with the unlabeled data in the USCL module and instructively generate the semantic-guided segmentation maps. Then multi-scale consistent constraints are further applied to the semantic-guided segmentation maps of each unlabeled image to provide detailed guidance on the entire learning process. In the test phase, both the above two modules can be dropped to maintain the model's simplicity. The main contributions are three-fold: (1) An effective and novel Inherent Consistent Learning framework is proposed to progressively learn robust category representations with semantic consistency between labeled and unlabeled data. (2) Two plug-and-play external modules, Supervised Semantic Proxy Adaptor (SSPA) and Unsupervised Semantic Consistent Learner (USCL), effectively leverage limited labeled data and massive unlabeled data only in the training phase. (3) We conduct experiments on three public benchmarks, and our proposed method outperforms the state-of-the-art by a margin, especially when the number of annotated data is extremely limited.

## 2. Methodology

For the semi-supervised segmentation setting, the training set usually contains a small portion of labeled set $D_l = \left\{x_i^l, y_i^l\right\}_{i=1}^{n}$ and a much larger scale unlabeled set $D_u = \left\{x_j^u\right\}_{j=1}^{m}$ (n $>>$ m). In this work, we propose the Inherent Consistent Learning (ICL) framework that can efficiently learn robust category representations on $D_l \cup D_u$ to enhance the segmentation precision with the guidance of a limited amount of $D_l$. Specifically, based on the attention mechanism, we construct a set of semantic-aware proxies that seek to preserve

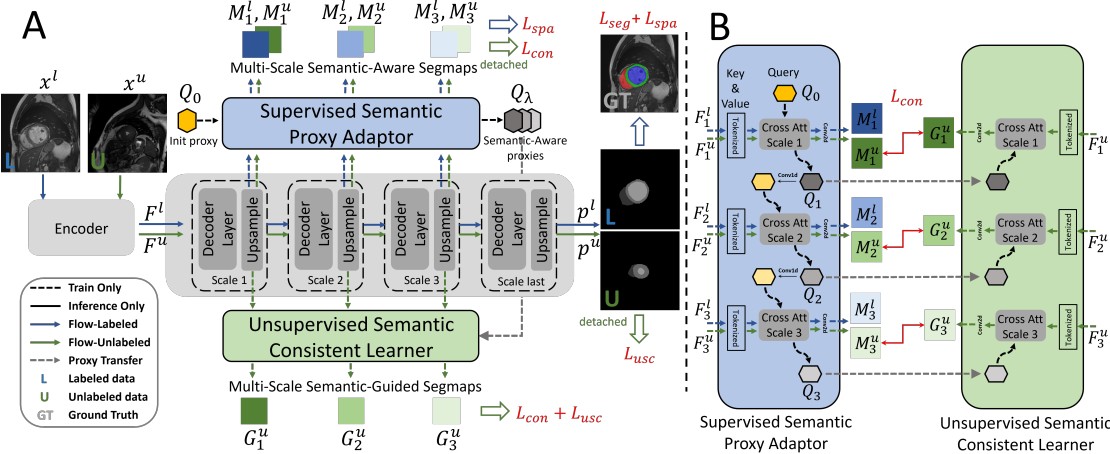

Figure 1: A: Overview of our proposed Inherent Consistent Learning framework. B: Supervised Semantic Proxy Adaptor and Unsupervised Semantic Consistent Learner.

the discriminability towards different categories. These semantic-aware proxies are trained with $D_l$ and then implicitly serve as a classifier to reveal the semantic-related areas within the unlabeled images at different representation levels.

As illustrated in Fig. 1, our framework consists of a single encoder-decoder model with a segmentation head, two newly proposed external modules named Supervised Semantic Proxy Adaptor (SSPA) and Unsupervised Semantic Consistent Learner (USCL), and an initialized semantic-aware proxy $Q \in \mathbb{R}^{Z \times 4C}$ that aims at transferring the well-learned semantic representation at different scales from limited labeled data. $Z$ and $C$ indicate the number of categories (including the background) and the tokenized feature dimension of the last scale. $4C$ is the dimension of the current proxy due to the scale factor. In training, both $x^l$ and $x^u$ are fed to the encoder-decoder backbone, generating multi-scale intermediate features $\{F^l, F^u\}$ for the SSPA and USCL modules to perform inherent consistent learning scheme by interacting with the initialized semantic-aware proxy. It is worth noting that such a strategy is suitable for both 2D and 3D networks, and these external modules are discarded during the inference phase.

## 2.1. Supervised Semantic Proxy Adaptor

We first define a learnable semantic-aware proxy as a query that aims to learn the statistics of organ categories (Xie et al., 2021). After that, we utilize the proxy to calculate the attention maps with the output features from the decoder. With a small convolutional head applied to attention maps, final predictions can be obtained. Later the well-trained proxy will be used to guide the learning of the USCL module.

For simplicity, 2D network is used for illustration. As depicted in Fig. 1B, a set of multi-scale intermediate features $\{F^l_\lambda, F^l_\lambda\}$ (scale $\lambda \in (1, 2, 3)$) are tokenized as $\{T^l_\lambda, T^u_\lambda\}$ and sent to a series of Cross-Attention Blocks within the SSPA module as input Key and Value. The input Query, i.e., initialized proxy $Q_0 \in \mathbb{R}^{Z \times 4C}$ is first sent to the cross-attention

module to interact with the tokenized features $\{T_1^l, T_1^u\} \in \mathbb{R}^{4C \times \left(\frac{H}{16} \times \frac{W}{16}\right)}$ at the first scale, generating the semantic-aware proxy $Q_1 \in \mathbb{R}^{Z \times 4C}$ and the semantic-aware attention maps $\{A_1^l, A_1^u\} \in \mathbb{R}^{Z \times \left(\frac{H}{16} \times \frac{W}{16}\right)}$ via Cross-Attention Mechanism (Details in Sec. A). $Q_1$ is then sent to the next cross-attention module after being applied a $1 \times 1$ convolution to reduce the dimension from 4C to 2C, obtaining the semantic-aware attention map at scale 2 and so on. At each scale the attention map $\{A_\lambda^l, A_\lambda^u\} \in \mathbb{R}^{Z \times \left(\frac{\lambda H}{16} \times \frac{\lambda W}{16}\right)}$ can be applied with a segmentation head to get the final predictions $\{M_\lambda^l, M_\lambda^u\} \in \mathbb{R}^{Z \times \frac{\lambda H}{16} \times \frac{\lambda W}{16}}$. To have strong discrimination for the semantic representations for different categories, we apply the cross-entropy and dice loss to actively supervise the $M_\lambda^l$ from the labeled data with $y^l$:

$$\mathcal{L}_{\text{spa}} = \frac{1}{\lambda} \sum_\lambda \left[ \mathcal{L}_{dice}\left(\sigma\left(\mathbb{I}\left(M_\lambda^l\right)\right), y^l\right) + \mathcal{L}_{ce}\left(\sigma\left(\mathbb{I}\left(M_\lambda^l\right)\right), y^l\right) \right], \tag{1}$$

where $\lambda \in (1, 2, 3)$ represents different scales, $\sigma(\cdot)$ denotes the softmax operation, and $\mathbb{I}(\cdot)$ is the bilinear interpolation to upsample the predictions to the same size as label $y^l$.

## 2.2. Unsupervised Semantic Consistent Learner

As depicted in Fig. 1B(right), the USCL module receives semantic-aware proxies $Q_\lambda \in \mathbb{R}^{Z \times \frac{4C}{\lambda}}$ with rich category semantic information from SSPA. They are utilized to reveal the semantic-related areas of the unlabeled intermediate features at different representation levels, instructively guiding the USCL module to generate the semantic-guided attention maps $\bar{A}_\lambda^u \in \mathbb{R}^{Z \times \left(\frac{\lambda H}{16} \times \frac{\lambda W}{16}\right)}$ at different scale $\lambda$. Similar to the SSPA module, these attention maps are used to get the guided segmentation maps $G_\lambda^u \in \mathbb{R}^{Z \times \frac{\lambda H}{16} \times \frac{\lambda W}{16}}$, but the USCL module only processes data from unlabeled source. By applying dice loss between the prediction $p^u$ of the network and $G_\lambda^u$, we can efficiently leverage the large amount of unlabeled data to boost the final segmentation performance of the main network:

$$\mathcal{L}_{\text{usc}} = \frac{1}{\lambda} \sum_\lambda \left[ \mathcal{L}_{dice}\left(\sigma\left(\mathbb{I}\left(G_\lambda^u\right)\right), \sigma\left(p^u\right)\right) \right] \tag{2}$$

where $p^u$ is the prediction with no gradient back-propagation from unlabeled data.

The learned proxies used to guide USCL are updated via SSPA with a new batch of samples. Considering the perturbation of the semantic-aware proxies introduced in this update process, inconsistency may exist in the segmentation maps of these two modules given the same unlabeled input sample. Different from the training strategy of SSPA, there is no ground-truth segmentation mask to supervise directly USCL. We use the above inconsistency to learn discriminative features from the unlabeled data. Specifically, we design a consistency regularization between $G_\lambda^u$ with $M_\lambda^u$ at different scales, which can leverage a large amount of unlabeled data to enhance the robustness of the global semantic representation. The Mean Squared Error (MSE) is adopted to form the loss as follow:

$$\mathcal{L}_{\text{con}} = \frac{1}{\lambda} \sum_\lambda \left[ \mathcal{L}_{MSE}\left(\sigma\left(G_\lambda^u\right), \sigma\left(M_\lambda^u\right)\right) \right] \tag{3}$$

Note that here the semantic-aware segmentation maps $M_\lambda^u \in \mathbb{R}^{Z \times \frac{\lambda H}{16} \times \frac{\lambda W}{16}}$ is detached to avoid disturbance of the well-trained SSPA.

## 2.3. The overall objective function

The goal of our inherent consistent learning (ICL) framework is to minimize the following combined objective function that contains the supervised and unsupervised parts:

$$\mathcal{L}_{total} = \underbrace{\mathcal{L}_{seg} + \mathcal{L}_{spa}}_{\text{supervised}} + \underbrace{\alpha\mathcal{L}_{usc} + \beta\mathcal{L}_{con}}_{\text{unsupervised}} \tag{4}$$

$$\mathcal{L}_{seg} = \mathcal{L}_{dice}\left(p^l, y^l\right) + \mathcal{L}_{ce}\left(p^l, y^l\right) \tag{5}$$

where $\mathcal{L}_{seg}$ is only used for supervising the final predictions of the labeled data, and we apply different hyper-parameters $\alpha, \beta$ for different datasets. (Details in Sec. B.3):

## 3. Experiments

### 3.1. Datasets and evaluation metrics

In this work, we evaluate our proposed method on **ACDC** (Bernard et al., 2018) and **AMOS** (Ji et al., 2022). (More experiments on **BraTS** (Menze et al., 2014) in Sec. C.2.)

**ACDC** dataset contains 200 cardiac cine-MR images with three segmentation targets, the left ventricle (LV), the myocardium (Myo) and the right ventricle (RV). For a fair comparison, we perform 2D segmentation and use the same data split, pre-processing and augmentations following (Luo et al., 2022a). Specifically, We use 2D slices from 3 and 7 cases (6/140 and 14/140 scans) to train and the remaining 60 scans to validate.

**AMOS** dataset provides 500 CT and 100 MRI scans, each with voxel-level annotations of 15 abdominal organs such as Spleen, Left and Right kidney, etc. In this work, we only use 300 CT scans from the AMOS 2022 official grand challenge (Ji et al., 2022), including 200 scans for training and 100 for validation. In detail, we divide the original validation set in a ratio of 3:7 to obtain the validation set and test set. Furthermore, 15/200 and 30/200 scans in the training set are used as two different semi-supervised settings.

During the validation phase, we perform slice-to-slice inference for 2D segmentation and stack them into a 3D prediction volume. For 3D segmentation, sliding window inference is applied to get the 3D predictions. To evaluate the segmentation results, we use the metrics 1) Dice Coefficient (DSC) and 2) 95% Hausdorff Distance (HD95).

### 3.2. Experimental details

**Network architectures.** Since we propose a plug-and-play scheme, we investigate the performance based on different networks, including CNN-Based networks for 2D and 3D segmentation, namely 2D- and 3D-UNet (Ronneberger et al., 2015). (More experiments on Transformer-Based (Trans-Based) networks of SwinUNet (Cao et al., 2021) and Swin-UNETR (Hatamizadeh et al., 2022) in Sec. C.1, C.2.)

**Comparison with baselines and existing methods.** We compare our proposed framework with four baselines and some recent SOTA semi-supervised segmentation methods, including Entropy Minimization (Ent-Mini) (Vu et al., 2019), Cross Consistency Training (CCT) (Ouali et al., 2020), FixMatch (Sohn et al., 2020), Regularized Dropout (R-Drop) (Wu et al., 2021), Cross Pseudo Supervision (CPS) (Chen et al., 2021b), Uncertainty Rectified Pyramid Consistency (URPC) (Luo et al., 2022b) and Cross Teaching between CNN

Table 1: Comparisons of the SOTA methods on ACDC using CNN-Based models. Mean and standard variance (in parentheses) are presented in the table.

| Labeled | CNN-Based | RV | | Myo | | LV | | Mean | |
|---|---|---|---|---|---|---|---|---|---|
| | | DSC ↑ | HD95 ↓ | DSC ↑ | HD95 ↓ | DSC ↑ | HD95 ↓ | DSC ↑ | HD95 ↓ |
| 3 cases | UNet[MICCAI15] | 41.26(33.74) | 79.78(125.91) | 56.73(26.44) | 29.21(80.07) | 65.04(29.25) | 32.78(80.81) | 54.34(29.81) | 47.25(95.59) |
| | Ent-Mini[CVPR19] | 46.56(29.91) | 39.86(32.72) | 61.46(24.02) | 29.37(67.14) | 71.74(25.72) | 33.11(68.69) | 59.92(26.55) | 34.11(56.18) |
| | CCT[CVPR20] | 46.88(31.96) | 29.65(52.68) | 63.18(22.92) | 21.14(24.49) | 68.88(26.47) | 33.56(31.22) | 59.65(27.12) | 28.11(36.13) |
| | FixMatch[NeurIPS20] | 66.98(26.62) | 28.22(67.68) | 71.3(18.61) | 9.22(21.52) | 81.73(19.77) | 9.03(20.07) | 73.34(21.67) | 15.49(36.42) |
| | R-Drop[NeurIPS21] | 43.55(34.04) | 97.72(152.73) | 63.46(25.02) | 13.23(47.62) | 72.93(25.95) | 26.94(80.48) | 59.98(28.34) | 45.96(93.62) |
| | CPS[CVPR21] | 49.46(33.21) | 43.93(90.65) | 63.24(24.14) | 18.9(66.64) | 74.96(24.53) | 28.01(67.59) | 62.56(28.29) | 30.28(74.96) |
| | URPC[MIA22] | 51.05(27.98) | 43.16(68.09) | 60.43(24.13) | 9.73(14.25) | 71.32(26.53) | 17.19(24.51) | 60.93(26.21) | 23.36(35.62) |
| | CTCT[MIDL22] | 71.68(26.24) | 28.96(92.18) | 70.58(23.82) | 9.28(47.58) | 80.23(23.56) | 16.55(66.77) | 74.16(24.54) | 18.27(68.85) |
| | **Ours** | **81.75(10.82)** | **3.56(4.52)** | **79.37(13.09)** | **4.93(8.34)** | **86.01(17.46)** | **6.86(7.89)** | **82.37(13.79)** | **4.28(6.92)** |
| 7 cases | UNet[MICCAI15] | 68.92(28.24) | 9.53(13.28) | 78.26(9.24) | 6.6(11.78) | 85.18(11.36) | 10.51(17.42) | 77.46(16.28) | 8.88(14.16) |
| | Ent-Mini[CVPR19] | 73.42(22.73) | 4.85(7.98) | 80.47(8.45) | 5.36(10.19) | 87.73(9.66) | 9.14(16.76) | 80.54(13.61) | 6.45(11.64)) |
| | CCT[CVPR20] | 80.08(15.95) | 5.85(10.4) | 82.85(7.63) | 5.36(10.4) | 89.17(9.13) | 13.97(21.93) | 84.01(10.9) | 8.4(14.24) |
| | FixMatch[NeurIPS20] | 84.22(15.05) | 2.01(2.50) | 83.14(6.38) | 2.5(5.73) | 89.70(9.68) | 6.08(12.67) | 85.69(10.37) | 3.53(6.97) |
| | R-Drop[NeurIPS21] | 72.7(26.22) | 5.24(7.64) | 81.38(6.79) | 4.4(8.83) | 89.28(8.42) | 7.18(16.32) | 81.12(13.81) | 5.24(10.93) |
| | CPS[CVPR21] | 80.47(17.55) | 3.51(6.75) | 82.65(6.38) | 6.36(11.54) | 87.89(10.03) | 11.39(17.98) | 83.67(11.32) | 7.09(12.09) |
| | URPC[MIA22] | 81.57(16.69) | 4.3(7.69) | 82.41(8.54) | 5.95(13.12) | 89.84(8.78) | 8.31(17.76) | 84.61(11.34) | 6.19(12.86) |
| | CTCT[MIDL22] | 85.37(10.54) | 3.45(8) | 84.77(4.81) | 7.05(18.04) | 90.21(8) | 9.2(16.2) | 86.78(7.78) | 5.43(10.58) |
| | **Ours** | **88.24(8.63)** | **1.67(1.46)** | **86.71(4.85)** | **1.6(1.94)** | **92(7.02)** | **3.54(6.88)** | **88.98(6.84)** | **2.27(3.43)** |
| Total | UNet[MICCAI15] | 90.07(7.53) | 1.31(0.78) | 88.87(3.3) | 1.09(0.37) | 94.32(3.8) | 1.52(3.6) | 91.09(4.88) | 1.31(1.581) |

and Transformer (CTCT) (Luo et al., 2022a). For a fair comparison, all implementations of these methods are consistent with our framework under the same task. All the auxiliary training modules are discarded, and only the trained backbones are used to generate final predictions. The above methods are openly available (Luo, 2020).

### 3.3. Results of 2D segmentation on ACDC dataset

**Improvements over baselines.** Tab.1 presents the results of all methods on the ACDC dataset using CNN-Based models (UNet). All methods have an improvement over the baseline under 3 and 7 labeled cases. Specifically, our proposed method surpasses the baseline by a large margin, with an average improvement of 28% in DSC and 43mm in HD95 under 3 labeled cases, 11.5% in DSC and 6.6mm in HD95 under 7 labeled cases. Especially for the category RV, our method improves the DSC from 41.26% to 81.75% and reduces the HD95 from 79.78mm to 3.56mm using only 3 labeled cases.

**Comparison with SOTA.** Compared with some SOTA methods, our approach outperforms the best under all partition protocols, especially when the number of labeled scans is extremely limited. In detail, Tab.1 shows that our method significantly surpasses the current SOTA method CTCT (Luo et al., 2022a) by 8.21% in DSC and 14mm in HD95 under 3 labeled cases. And we also achieved great improvement in DSC and HD95 (2.2% larger and 3.16mm smaller than CTCT) using only 7 labeled cases. The first three columns in Fig. 2(left) present a visualization of the segmentation performance of different methods. Compared with others, our method presents more stable results for hard-to-segment categories like the RV and even stronger discrimination for the background and the foreground categories when they have similar and ambiguous representations.

### 3.4. Results of 3D segmentation on AMOS dataset

**Improvements over baselines.** Tab.2 reports partial quantitative comparison results of our method and others under 15/200 and 30/200 labeled settings (Complete results in

Table 2: Comparisons of the SOTA methods on AMOS using CNN-Based models. Full results of all categories are presented in Tab. 11. Mean and standard variance (in parentheses) are presented in the table.

| | | DSC ↑ | | | | | | | | | | |
|---|---|---|---|---|---|---|---|---|---|---|---|---|
| Labeled | CNN-Based | Large | | | Medium | | | | Small | | | |
| | | SPL | LIV | STO | BLA | PAN | IVC | DUO | GBL | RAG | LAG | Mean |
| 15 scans | 3D-UNet[MICCAI15] | 14.98(13.84) | 73.37(11.66) | 19.02(17.17) | 15.38(16.27) | 20.94(17.44) | 58.01(16.92) | 14.69(12.83) | 7.44(17.9) | 15.98(15.32) | 13.36(20.14) | 31.34(17.6) |
| | Ent-Mini[CVPR19] | 76.26(17.08) | 80.94(9.41) | 48.64(19.49) | 42.08(21.34) | 30.61(16.9) | 59.25(16.2) | 15.84(14.44) | 23.32(22.74) | 29.78(20.19) | 24.58(23.33) | 49.76(18.74) |
| | CTCT[MIDL22] | 73.48(12.15) | 70.13(10.07) | 39.16(17.76) | 27.79(20.76) | 22.49(14.31) | 50.92(12.4) | 13.72(12.91) | 4.29(20.25) | 13.04(14.89) | 23.04(19.99) | 40.82(16.08) |
| | **Ours** | **88.11(12.38)** | **89.45(6.92)** | **60.29(22.28)** | **51.41(27.71)** | **62.47(16.86)** | **69.98(11.99)** | **42.54(19.1)** | **40.26(29.7)** | **44.7(21.21)** | **40.21(25.21)** | **61.44(19.5)** |
| 30 scans | 3D-UNet[MICCAI15] | 71.83(21.51) | 90.94(8.88) | 68.14(23.86) | 62.81(25.46) | 67.99(16.48) | 81.95(7.96) | 50.74(20.28) | **65.05(30.36)** | 57.11(17.37) | 56.03(21.07) | 68.47(18.59) |
| | Ent-Mini[CVPR19] | 83.21(15.77) | 90.07(8.44) | 62.92(25.31) | 52.02(28.86) | 41.45(22.45) | 72.78(11.7) | 42(20.39) | 41.64(30.19) | 47.83(22.61) | 29.79(23.21) | 59.74(20.91) |
| | CTCT[MIDL22] | **89.39(7.67)** | **93.32(5.51)** | 65.95(22.58) | 55.21(23.32) | 65.58(14.33) | 77.12(8.4) | 52.03(15.96) | 63.38(25.03) | **64.74(13.96)** | 46.52(22.6) | 70.41(15.64) |
| | **Ours** | 81.63(15.2) | 92.24(7.3) | **75.01(20.85)** | **67.81(27.34)** | **70.34(15.67)** | **83.78(6.84)** | **58.56(19.77)** | 64.5(29.43) | 61.15(16.31) | **58.44(20.12)** | **74.81(16.56)** |
| Total | 3D-UNet[MICCAI15] | 94.21(5.94) | 96.43(2.27) | 87.89(15.81) | 83.53(18.9) | 83.27(11.51) | 88.02(4.78) | 77.43(14.4) | 80.61(23.38) | 71.83(13.35) | 73.43(13.45) | 85.17(11.71) |

| | | HD95 ↓ | | | | | | | | | | |
|---|---|---|---|---|---|---|---|---|---|---|---|---|
| Labeled | CNN-Based | Large | | | Medium | | | | Small | | | |
| | | SPL | LIV | STO | BLA | PAN | IVC | DUO | GBL | RAG | LAG | Mean |
| 15 scans | 3D-UNet[MICCAI15] | 39.69(31.09) | 17.99(16.85) | 64.2(62.24) | 41.39(62.47) | 27.14(19.27) | 15.46(27.51) | 41.36(52.49) | 116.4(153) | 79.58(118) | 155.9(177.5) | 51.39(63.97) |
| | Ent-Mini[CVPR19] | 13.58(28.33) | 13.89(24.21) | 50.1(72.23) | **17.57(43.32)** | 21.29(43.16) | 9.26(15.06) | 26.44(43.58) | 55.83(86.59) | **18.6(47.28)** | 87.21(149.65) | 25.99(48.5) |
| | CTCT[MIDL22] | 33.98(42.47) | 11.11(11.52) | 37.45(52.03) | 25.17(43.8) | 36.52(31.28) | 73.3(23.62) | 39.49(45.24) | 357.14(75.57) | 108.05(161.82) | **35.74(71.12)** | 63.98(56.06) |
| | **Ours** | **9.13(23.78)** | **9.33(21.83)** | **22.49(47.53)** | 19.83(47.56) | **10.41(21.27)** | **6.44(8.38)** | **16.75(43.54)** | **33.91(86.06)** | 23.78(74.81) | 52.56(123.2) | **17.85(43.27)** |
| 30 scans | 3D-UNet[MICCAI15] | 10.57(19.29) | 10.88(25.58) | 15.5(44.2) | **7.62(6.41)** | 8.38(18.39) | 2.91(4.49) | 15.44(44.77) | 32.95(95.03) | 3.73(2.61) | 11.75(45.95) | 13.44(31.07) |
| | Ent-Mini[CVPR19] | 16.24(32.13) | 7.87(18.29) | 17.99(44.53) | 16.5(43.88) | 15.73(17.97) | 3.98(2.8) | 17.96(43.63) | 31.86(84.70) | 15.31(61.48) | 44.01(109.77) | 18.8(43.37) |
| | CTCT[MIDL22] | 7.58(21.85) | 4.61(17.33) | 15.38(44.18) | 16.6(44.04) | 7.25(7.04) | 3.23(2.57) | 13.96(43.52) | 23.45(76.08) | **2.81(2.1)** | 10.87(43.75) | 10.34(30.2) |
| | **Ours** | 7.77(15.91) | 9.76(23.93) | 13.5(44.51) | 8.19(11.06) | **5.32(5.44)** | **1.91(1.29)** | 13.46(43.96) | 26.22(85.68) | 3.41(3.26) | **8.89(43.91)** | **9.87(28.68)** |
| Total | 3D-UNet[MICCAI15] | 0.77(2.18) | 0.26(1.23) | 8.2(44.33) | 3.91(9.8) | 2.49(4.65) | 1.34(0.81) | 8.83(43.97) | 13.73(62.01) | 7.28(44.07) | 2.32(3.23) | 4.83(21.94) |

Tab. 11). The results in Tab.2 show that our method exceeds the baselines by a large margin, especially under 15/200 protocol, achieving the tremendous improvement of over 30% in mean DSC and 33.5mm in mean HD95. When using 30 labeled scans, there is still an obvious improvement over the baseline, with a gain of 6.3% in mean DSC and 3.6mm in mean HD95. Although all methods show enhancement under 15 labeled scans, Ent-Mini gets degraded results when using 30 labeled scans.

**Comparison with SOTA.** Here we conduct comparative experiments on two existing methods considering the training cost and resource limitation. As presented in Tab.2, our proposed method attains mean DSC and HD95 of 61.44% and 17.85mm under 15 labeled scans, 74.81% and 9.87mm under 30 labeled scans, which outperforms the SOTA methods Ent-Mini and CTCT by a large margin. Particularly, under the least labeled data setting (15/200), our method surpasses the Ent-Mini and CTCT with improvements of 11.7% and 20.6% in DSC, 8.1mm and 46.1mm in HD95, respectively. We also notice that our proposed method has remarkable performance when segmenting medium and small-size organs that are hard to distinguish, especially with limited labeled training data. The visual comparison base on 30 labeled data is presented in the last four columns in Fig. 2(left), It shows that our method can generate better results with fewer false-positive regions, especially on medium-size and small-size organs.

### 3.5. Ablation study:

**Analysis of the discrimination ability towards the semantic-aware proxy.** In this section, we produce qualitative analysis to demonstrate the semantic-aware proxy can distinguish the semantic representation of different categories well. In detail, we utilize the well-learned proxy to interact with all labeled data successively in the first scale of the SSPA module. By regarding each updated proxy as a sample, we apply the t-SNE (Van der Maaten and Hinton, 2008) to visualize the proxy distribution of different categories in ACDC and AMOS datasets. In Fig.2(right), different categories are clearly separated with a larger metric distance, proving the discriminability of our semantic-aware proxy.

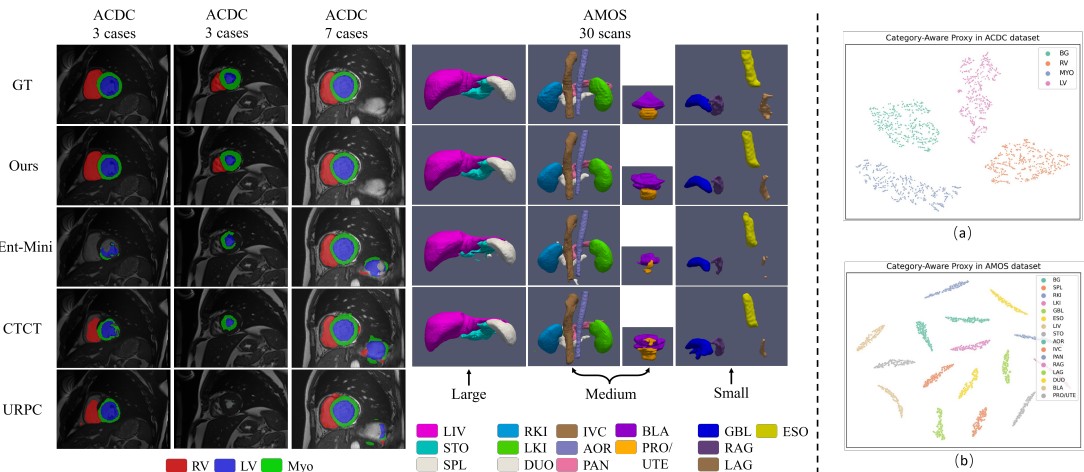

Figure 2: Left: Visual comparisons on ACDC and AMOS (zoom in for better observation). Right: Visual results of the t-SNE on the semantic-aware proxy.

**Analysis of the Supervised Semantic Proxy Adaptor.** In this section, we conduct experiments on the ACDC dataset using 7 labeled cases and without unlabeled data to further investigate the effectiveness of the Supervised Semantic Proxy Adaptor. Particularly, we utilize UNet and SwinUNet as our backbone modules. As presented in Tab. 3, our proposed SSPA module brings promising gains to the segmentation results in overall categories on the CNN-Based and Trans-Based networks. Specifically, the SSPA module improves 5% in the mean DSC for UNet, boosts 2.5% in the mean DSC, and reduces 3.7mm in HD95 for SwinUNet. More experiments can be found in Sec. C.3.

Table 3: Effects of Supervised Semantic Proxy Adaptor on ACDC dataset.

| Labeled | 2D Models | RV | | Myo | | LV | | Mean | |
|---|---|---|---|---|---|---|---|---|---|
| | | DSC ↑ | HD95 ↓ | DSC ↑ | HD95 ↓ | DSC ↑ | HD95 ↓ | DSC ↑ | HD95 ↓ |
| 7 cases | UNet[MICCAI15] | 68.92(28.24) | 9.53(13.28) | 78.26(9.24) | 6.6(11.78) | 85.18(11.36) | 10.51(17.42) | 77.46(16.28) | 8.88(14.16) |
| | UNet w/ SSPA | 77.59(20.58) | 5.9(9.71) | 81.87(5.99) | 8.08(15.89) | 88.38(7.99) | 11.16(18.94) | 82.61(11.52) | 8.38(14.85) |
| | SwinUNet[arXiv21] | 75.17(16.72) | 10.74(21.86) | 74.69(9.65) | 4.75(6.9) | 83.13(14.47) | 9.6(13.36) | 77.75(13.61) | 8.36(14.04) |
| | SwinUNet w/ SSPA | 78.03(16.36) | 2.46(2.11) | 77.77(7.96) | 4.17(6.55) | 85.03(13.04) | 7.39(10.28) | 80.27(12.45) | 4.67(6.31) |

## 4. Conclusion

In this paper, we propose Inherent Consistent Learning (ICL) framework that aims to progressively learn robust category representations from scarce labeled data and numerous unlabeled data to help segmentation under the semi-supervised scenario. To achieve this goal, we introduce two plug-and-play modules, namely Supervised Semantic Proxy Adaptor (SSPA) and Unsupervised Semantic Consistent Learner, based on attention mechanism to align the semantic representations of labeled and unlabeled data, providing semantic consistency guidance to boost the final segmentation performance. Experimental results on three open benchmarks demonstrate the effectiveness of the proposed method. Future work may focus on semi-supervised domain adaption problems adopting the proposed framework.

## Acknowledgments

This work is supported in part by Chinese Key-Area Research and Development Program of Guangdong Province under grant No. 2020B0101350001, by Young Scientists Fund of the National Natural Science Foundation of China under grant No. 62106154, by Natural Science Foundation of Guangdong Province, China (General Program) under grant No.2022A1515011524, by Shenzhen Science and Technology Program ZDSYS20211021111415025, and by the Guangdong Provincial Key Laboratory of Big Data Computing, The Chinese Universi of Hong Kong (Shenzhen).

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

## Appendix A. Methodology details

### A.1. Semantic-aware Proxy Progressive Updating

In Sec. 2.1, the initialized proxy is sent to the cross-attention module to conduct interaction with the tokenized features $\left\{T_1^l, T_1^u\right\} \in \mathbb{R}^{4C \times \left(\frac{H}{16} \times \frac{W}{16}\right)}$ at the first scale, generating the semantic-aware proxy $\left\{Q_1^l, Q_1^u\right\} \in \mathbb{R}^{Z \times 4C}$ and the semantic-aware attention maps $\left\{A_1^l, A_1^u\right\} \in \mathbb{R}^{Z \times \left(\frac{H}{16} \times \frac{W}{16}\right)}$. Progressively, the updated proxy is then sent to the next scale of cross-attention after being applied a $1 \times 1$ convolution to reduce the dimension from 4C to 2C, obtaining the semantic-aware attention maps at scale 2 and so on.

For simplicity, we illustrate the cross-attention (CA) updating process using the tokenized features $T_1^l$ at scale 1 and the initialized proxy $Q$ as follows,

$$q = QW_Q, \ k = T_1^l W_K, \ v = T_1^l W_V, \tag{6}$$

$$\mathrm{CA}(Q, T_1^l) = \mathrm{softmax}(\frac{qk^\mathsf{T}}{\sqrt{d}})v, \tag{7}$$

where $W_K$, $W_V$, $W_Q \in \mathbb{R}^{4C \times 4C'}$ are the parameter matrices for linear projection. Here $d$ is the channel dimension $4C$. The softmax($\cdot$) denotes the softmax function along the spatial dimension. The $qk^{\mathsf{T}} \in \mathbb{R}^{Z \times \frac{H}{16} \times \frac{W}{16}}$ indicates the *Semantic-aware Feature Maps* extracted from a single CA head, where $Z$ denotes the total numbers of classes. The Multi-head Cross Attention (MCA) is the extension with $N$ independent CAs and projects their concatenated outputs as follows:

$$\text{MCA}(Q, T_1^l) = \mathbb{C}(\text{ CA}_1(Q, T_1^l), ..., \text{CA}_N(Q, T_1^l) \text{ }) W_O, \tag{8}$$

where $\mathbb{C}$ denotes the concatenation operation. $W_O \in \mathbb{R}^{4C' \times 4C}$ is the learnable parameter matrix, and we have $4C' = 4C/N$. Here, we can obtain the semantic-aware attention maps $A_1^l \in \mathbb{R}^{Z \times (\frac{H}{16} \times \frac{W}{16})}$ extracted from multi-heads. Finally, the $Q$ can be updated by:

$$\hat{Q}_1^l = \text{MCA}(\text{ Norm}(Q), \text{ Norm}(T_1^l) \text{ }) + Q, \tag{9}$$

$$Q_1^l = \text{Conv}(\text{MLP}(\text{ Norm}(\hat{Q}_1^l) \text{ }) + \hat{Q}_1^l), \tag{10}$$

where $Q_1^l$ indicates the updated semantic-aware proxy at scale 1. Conv is a $1 \times 1$ convolutional layer to reduce the channel dimension from 4C to 2C. The operations with the tokenized features $T_1^u$ at scale 1 are identical to those described above. **It is worth noting** that $T_1^u$ has no effect on the updating of the learnable parameter $Q$ during back-propagation, while $T_1^l$ does. Moreover, the other scales are also the same as the above equations, and the only difference is that we adopt the semantic-aware proxy (e.g. $Q_1^l$ or $Q_2^l$) to replace $Q$.

## Appendix B. Experimental details

### B.1. BraTS dataset

**BraTS** dataset includes 335 MR scans from four different modalities (FLARE, T1, T1ce and T2). Here we conduct 3D segmentation using the whole tumors from FLARE (250/335) following (Luo et al., 2022b), utilizing the same data split, augmentations and pre-processing strategy. For the semi-supervised learning setting, we use 25/250 and 50/250 scans as different portions of the training set and 85 scans as the validation set.

### B.2. AMOS dataset

#### B.2.1. Category details.

AMOS dataset aims to promote abdominal multi-organ segmentation under diverse, complex and clinical scenarios. To get a clearer vision of the effectiveness of our proposed method, we objectively divide the abdominal organs into three different sizes as follows:

- **Large**: spleen (SPL), liver (LIV) and stomach (STO)

- **Medium**: right kidney (RKI), left kidney(LKI), bladder (BLA), aorta (AOR), pancreas (PAN), infer vena cava (IVC), duodenum (DUO) and prostate/uterus (PRO/UTE)

- **Small**: gallbladder (GBL), esophagus (ESO), right adrenal gland (RAG) and left adrenal gland (LAG)

B.2.2. Data splits.

In the AMOS dataset, 300 CT scans are derived from three distinct scanners (domains) to ensure data diversity, including 200 scans for training and 100 for validation. In particular, we randomly selected one sample of each domain from the training set and repeated 5 times, obtaining 15 labeled scans for the first semi-supervised setting. To get the other semi-supervised setting, we repeated the selection 5 more times and got 30 labeled scans with the former acquiring 15 labeled scans. Furthermore, we perform the similar strategy to choose 30 scans to form the validation set to validate and select the best model during training. The remaining 70 scans are used for later testing.

## B.3. Training details.

Our experiments are implemented in Pytorch (Paszke et al., 2019) using an NVIDIA A100 GPU. All the networks, including 2D and 3D, are trained by an SGD optimizer with the poly-learning rate strategy. Specifically, during training, a batch of data consists of half-labeled and half-unlabeled data.

**2D networks.** For the ACDC dataset, we train the 2D models for 30000 iterations with an initial learning rate (lr) of 0.01 and a batch size of 16, validate the models every 200 iterations, and find the best models to make the final comparison.

**3D networks.** For the BraTS dataset, we set the batch size to 4 with an initial lr of 0.01 and train all 3D models for 30000 iterations. Considering that the AMOS22 dataset with multiple categories may take a longer time for the models to converge, we train the models on AMOS dataset for 60000 iterations with a batch size of 4 and an initial lr of 0.02. To find the best model of each model, we validate the models every 200 (on BraTS) and 1200 (on AMOS) iterations.

As mentioned in Sec. 2.3, the unsupervised part of the overall objection function contains two hyper-parameters $\alpha$, $\beta$. During training, we set these hyper-parameters as (1, 50), (1, 10) and (0.1, 10) for ACDC, BraTs and AMOS datasets respectively.

## B.4. Computational cost.

Tab. 4 reports the total training time and per scan inference time of different SOTA methods and our proposed method under 7 labeled cases in the ACDC dataset using the same device. It can be found that our method needs longer training costs due to the attention-mechanism-based modules that we proposed. But all the methods have a similar time since we utilize the same backbone network to perform inference.

Table 4: Comparison of the computational cost of the different SOTA methods and our proposed method under 7 labeled cases on the ACDC dataset. TTimes (hours) denotes the total training time, and ITimes (seconds) denotes the inference time per scan.

|  | UNet | Ent-Mini | CCT | FixMatch | R-Drop | CPS | URPC | CTCT | **Ours** |
|---|---|---|---|---|---|---|---|---|---|
| TTimes (h) | 1.03 | 1.56 | 1.55 | 1.16 | 1.14 | 1.03 | 0.77 | 1.72 | **2.46** |
| ITimes (s) | 0.18 | 0.18 | 0.18 | 0.18 | 0.18 | 0.18 | 0.18 | 0.18 | **0.18** |

## Appendix C. Additional experiments.

To investigate the generalization ability of our proposed method, we further conduct additional experiments adopting Trans-Based models on the ACDC and both Cnn-Based and Trans-Based for the BraTS dataset. And these additional experimental results further demonstrate the generality of our proposed method. Besides, we present the full results of all categories in the AMOS dataset based on CNN-Based as a complement in Tab. 11.

### C.1. Experiments of ACDC dataset on Trans-Based models.

**Improvements over baselines.** As shown in Tab. 5, our method still has tremendous improvements over the baseline using Trans-Based backbone by 15.1% in mean DSC and -6.2mm in mean HD95 under 3 labeled cases, yet some of the compared approaches (Ent-Mini, R-Drop and CPS) suffer from declined performance. **Comparison with SOTA.** Compared with the current SOTA method using the Trans-Based model, we surpass the CTCT by 3.1% in mean DSC and 4.5mm in mean HD95 under 3 labeled cases, 2% in mean DSC and 2.4mm in mean HD95. Our method significantly affects the metric HD95, where we achieve 1.91mm, extremely close to the fully supervised baseline performance (1.56mm).

Table 5: Comparisons of the SOTA methods on the ACDC using Trans-Based models. Mean and standard variance (in parentheses) are presented in the table.

| Labeled | Trans-Based | RV | | Myo | | LV | | Mean | |
|---|---|---|---|---|---|---|---|---|---|
| | | DSC ↑ | HD95 ↓ | DSC ↑ | HD95 ↓ | DSC ↑ | HD95 ↓ | DSC ↑ | HD95 ↓ |
| 3 cases | SwinUNet[arXiv21] | 58.53(27.06) | 10.15(15.66) | 58.69(21.84) | 8.03(9.16) | 71.63(24.86) | 10.69(11.23) | 62.95(24.58) | 9.62(12.02) |
| | Ent-Mini[CVPR19] | 29.93(22.24) | 16.9(11.82) | 36.78(18.34) | 20.01(16.31) | 48.92(24.6) | 27.53(19.72) | 38.54(21.73) | 21.48(15.95) |
| | R-Drop[NeurIPS21] | 30.57(21.56) | 24.82(21.27) | 36.11(19.4) | 18.31(16) | 48.59(24.78) | 23.67(15.33) | 38.43(21.91) | 22.27(17.53) |
| | CPS[CVPR21] | 30.51(21.58) | 24.88(21.33) | 36.13(19.41) | 18.38(16.03) | 48.63(24.76) | 23.81(15.41) | 38.43(21.92) | 22.36(17.59) |
| | CTCT[MIDL22] | 72.99(22.93) | 14.67(49.73) | 70.75(20.14) | 4.16(5.06) | 81.15(20.52) | 4.97(7.86) | 74.96(21.2) | 7.93(20.89) |
| | **Ours** | **76.85(14.21)** | **4.07(5.26)** | **72.31(16.36)** | **3.19(4.51)** | **85(14.42)** | **3.09(3.28)** | **78.06(15)** | **3.45(4.35)** |
| 7 cases | SwinUNet[arXiv21] | 75.17(16.72) | 10.74(21.86) | 74.69(9.65) | 4.75(6.9) | 83.13(14.47) | 9.6(13.36) | 77.75(13.61) | 8.36(14.04) |
| | Ent-Mini[CVPR19] | 44.15(22.17) | 21.36(22.58) | 50.46(17.72) | 12.49(11.46) | 62.64(21.32) | 18.28(13.15) | 52.42(20.4) | 17.38(15.73) |
| | R-Drop[NeurIPS21] | 48.42(20.77) | 19.84(20.13) | 49.58(17.46) | 10.80(11.11) | 63.81(21.9) | 16.32(12.44) | 53.94(20.04) | 15.65(14.56) |
| | CPS[CVPR21] | 46.82(21.02) | 23.64(22.35) | 50.55(17.2) | 11.8(11.59) | 63.24(21.53) | 17.40(12.38) | 53.54(19.92) | 17.61(15.44) |
| | CTCT[MIDL22] | 85.09(8.51) | 2.86(7.98) | 82.68(5.16) | 2.69(4.97) | 88.68(9.73) | 7.51(13.74) | 85.48(7.8) | 4.35(8.9) |
| | **Ours** | **85.78(10.37)** | **1.86(1.63)** | **84.94(4.28)** | **1.8(4.57)** | **91.88(5.82)** | **2.08(5.63)** | **87.54(6.83)** | **1.91(3.94)** |
| Total | SwinUNet[arXiv21] | 89.36(7.5) | 1.82(3.9) | 87.49(4.04) | 1.16(0.75) | 93.4(4.15) | 1.69(3.13) | 90.08(5.23) | 1.56(2.59) |

### C.2. Experiments on BraTS dataset on both CNN- and Trans-Based models.

**Improvements over baselines.** For the BraTS Dataset, the quantitative comparisons are shown in Tab. 6. It can be found that our method has significant improvements over the baselines under 25 and 50 annotated scans. Specifically, our CNN-Based method outperforms the 3D-UNet with a gain of 2.23% in DSC and 3.24mm in HD95 under 25 labeled cases and with a gain of 2.13% in DSC and 4.16mm in HD95 under 50 labeled cases. Our method using only 20% labeled scans has even better results than the fully supervised baseline using 250 labeled scans in HD95, which is 7.44mm and 7.52mm, respectively.

**Comparison with SOTA.** For the BraTS dataset, we compare our method with some recent semi-supervised segmentation methods. As observed, our proposed method outperforms all the compared methods with the highest DSC and lowest HD95 in all settings, including different partition protocols and backbones. In particular, we surpass the current best method URPC by a notable margin, achieving improvements of +2.55mm and

+2.31mm on the metric HD95 under 25/250 and 50/250 partition protocols, respectively. Fig.5 presents several tumor segmentation results of the compared methods and our proposed method. As observed, our method can generate better results with fewer false-positive regions.

Table 6: Comparisons of the SOTA methods on the BraTS using CNN- and Trans-Based models. In particular, the upper-bound (250 scans) performance of 3D-UNet is 86.77%(11.14) in DSC and 7.52mm(10.58) in HD95, and for SwinUNETR is 87.33%(10) in DSC and 7.24mm(9.6) in HD95. Mean and standard variance (in parentheses) are presented in the table.

| CNN-Based | Tumor (25 scans) | | Tumor (50 scans) | |
|---|---|---|---|---|
| | DSC ↑ | HD95 ↓ | DSC ↑ | HD95 ↓ |
| 3D-UNet[MICCAI15] | 82.67(12.8) | 13.96(19.05) | 83.76(12.27) | 11.6(16.4) |
| Ent-Mini[CVPR19] | 83.17(12.53) | 12.01(17) | 84.32(12.17) | 10.59(14.81) |
| R-Drop[NeurIPS21] | 83.21(11.46) | 13.05(15.96) | 84.41(11.7) | 9.91(13.08) |
| CPS[CVPR21] | 83.62(11.6) | 14.3(20.22) | 84.26(12.43) | 10.13(14.83) |
| CTCT[MIDL22] | 83.9(11.92) | 12.32(18.24) | 84.32(12.41) | 10.01(14.72) |
| URPC[MIA22] | 84.15(10.66) | 13.27(19.4) | 85.63(10.05) | 9.75(13.1) |
| **Ours** | **84.9(10.23)** | **10.72(14.02)** | **85.89(10.16)** | **7.44(8.65)** |
| Trans-Based | DSC ↑ | HD95 ↓ | DSC ↑ | HD95 ↓ |
| SwinUNETR[arXiv21] | 83.9(12.3) | 9.2(11.25) | 84.87(11.57) | 9.29(12.67) |
| Ent-Mini[CVPR19] | 84.09(12.24) | 10.67(13.84) | 84.97(11.72) | 8.29(10.28) |
| R-Drop[NeurIPS21] | 84.05(11.76) | 9.72(12.38) | 84.83(10.91) | 10.53(16.92) |
| CPS[CVPR21] | 83.79(12.52) | 10.02(12.51) | 84.99(11.16) | 9.22(11.51) |
| CTCT[MIDL22] | 84.55(11.25) | 10.21(13.12) | 85.08(11.47) | 9.65(13.17) |
| **Ours** | **85.38(10.71)** | **9.05(11.69)** | **85.63(10.93)** | **8.05(10.36)** |

Table 7: Effects of Supervised Semantic Proxy Adaptor on the ACDC dataset. Mean and standard variance (in parentheses) are presented in the table.

| Labeled | 2D Models | RV | | Myo | | LV | | Mean | |
|---|---|---|---|---|---|---|---|---|---|
| | | DSC ↑ | HD95 ↓ | DSC ↑ | HD95 ↓ | DSC ↑ | HD95 ↓ | DSC ↑ | HD95 ↓ |
| 3 cases | UNet[MICCAI15] | 41.26(33.74) | 79.78(125.91) | 56.73(26.44) | 29.21(80.07) | 65.04(29.25) | 32.78(80.81) | 54.34(29.81) | 47.25(95.59) |
| | UNet w/ SSPA | 38.13(32.27) | 104.38(149.63) | 61.80(26.26) | 22.71(67.03) | 70.99(28.05) | 20.14(51.33) | 56.97(28.86) | 49.08(89.33) |
| | SwinUNet[arXiv21] | 58.53(27.06) | 10.15(15.66) | 58.69(21.84) | 8.03(9.16) | 71.63(24.86) | 10.69(11.23) | 62.95(24.58) | 9.62(12.02) |
| | SwinUNet w/ SSPA | 57.51(27.35) | 7.38(13.27) | 60.56(21.94) | 8.5(9.86) | 71.09(25.7) | 11.17(11.49) | 63.05(24.99) | 9.01(11.54) |
| 7 cases | UNet[MICCAI15] | 68.92(28.24) | 9.53(13.28) | 78.26(9.24) | 6.6(11.78) | 85.18(11.36) | 10.51(17.42) | 77.46(16.28) | 8.88(14.16) |
| | UNet w/ SSPA | 77.59(20.58) | 5.9(9.71) | 81.87(5.99) | 8.08(15.89) | 88.38(7.99) | 11.16(18.94) | 82.61(11.52) | 8.38(14.85) |
| | SwinUNet[arXiv21] | 75.17(16.72) | 10.74(21.86) | 74.69(9.65) | 4.75(6.9) | 83.13(14.47) | 9.6(13.36) | 77.75(13.61) | 8.36(14.04) |
| | SwinUNet w/ SSPA | 78.03(16.36) | 2.46(2.11) | 77.77(7.96) | 4.17(6.55) | 85.03(13.04) | 7.39(10.28) | 80.27(12.45) | 4.67(6.31) |
| Total | UNet[MICCAI15] | 90.07(7.53) | 1.31(0.78) | 88.87(3.3) | 1.09(0.37) | 94.32(3.8) | 1.52(3.6) | 91.09(4.88) | 1.31(1.58) |
| | UNet w/ SSPA | 90.53(7) | 1.23(0.71) | 89.04(3.34) | 1.57(2.88) | 94.27(4.07) | 1.04(0.19) | 91.28(4.81) | 1.28(1.26) |
| | SwinUNet[arXiv21] | 89.36(7.5) | 1.82(3.9) | 87.49(4.04) | 1.16(0.75) | 93.4(4.15) | 1.69(3.13) | 90.08(5.23) | 1.56(2.59) |
| | SwinUNet w/ SSPA | 90.13(7.32) | 1.24(0.63) | 88.27(3.22) | 1.04(0.12) | 93.89(4) | 1.08(0.27) | 90.76(4.84) | 1.12(0.33) |

### C.3. Effects of the Supervised Semantic Proxy Adaptor.

In this section, we conduct experiments on ACDC and AMOS datasets to further investigate the effectiveness of the Supervised Semantic Proxy Adaptor module. In practice, we utilize the SSPA module with only labeled data to train the network, achieving surprising results. Tab. 7 and Tab. 8 show that our proposed SSPA module can take full advantage of the high-level semantic information and evidently improve the segmentation performance for

Table 8: Effects of Supervised Semantic Proxy Adaptor on the AMOS dataset. Mean and standard variance (in parentheses) are presented in the table.

| | | DSC ↑ | | | | | | | | | | |
|---|---|---|---|---|---|---|---|---|---|---|---|---|
| | | Large | | | Medium | | | | Small | | | |
| Labeled | CNN-Based | SPL | LIV | STO | BLA | PAN | IVC | DUO | GBL | RAG | LAG | Mean |
| 15 scans | 3D-UNet[MICCAI15] | 14.98(13.84) | 73.37(11.66) | 19.02(17.17) | 15.38(16.27) | 20.94(17.44) | 58.01(16.92) | 14.69(12.83) | 7.44(17.9) | 15.98(15.32) | 13.36(20.14) | 31.34(17.6) |
| | 3D-UNet w/ SSPA | 67.76(23.72) | 70.4(21.93) | 37.25(23.38) | 15.83(19.14) | 44.21(19.35) | 72.6(10.71) | 30.08(18.46) | 36.01(27.48) | 35.52(22.66) | 22.54(23.59) | 46.71(20.58) |
| 30 scans | 3D-UNet[MICCAI15] | 71.83(21.51) | 90.94(8.88) | 68.14(23.86) | 62.81(25.46) | 67.99(16.48) | 81.95(7.96) | 50.74(20.28) | 65.05(30.36) | 57.11(17.37) | 56.03(21.07) | 68.47(18.59) |
| | 3D-UNet w/ SSPA | 83.56(15.42) | 90.42(9.28) | 72.24(22.23) | 70.35(25.81) | 66.08(18.38) | 82.1(7.24) | 44.78(19.08) | 70.35(25.81) | 62.05(16.77) | 51.23(23.66) | 72.31(17.41) |
| Total | 3D-UNet[MICCAI15] | 94.21(5.94) | 96.43(2.27) | 87.89(15.81) | 83.53(18.9) | 83.27(11.51) | 88.02(4.78) | 77.43(14.4) | 80.61(23.38) | 71.83(13.35) | 73.43(13.45) | 85.17(11.71) |
| | 3D-UNet w/ SSPA | 94.18(6.73) | 96.77(1.44) | 86.84(17.51) | 84.76(17.91) | 83.08(11.44) | 88.61(4.65) | 76.77(15.66) | 81.62(20.8) | 68.63(11.99) | 70.98(11.66) | 84.92(11.09) |
| | | HD95 ↓ | | | | | | | | | | |
| | | Large | | | Medium | | | | Small | | | |
| Labeled | CNN-Based | SPL | LIV | STO | BLA | PAN | IVC | DUO | GBL | RAG | LAG | Mean |
| 15 scans | 3D-UNet[MICCAI15] | 39.69(31.09) | 17.99(16.85) | 64.2(62.24) | 41.39(62.47) | 27.14(19.27) | 15.46(27.51) | 41.36(52.49) | 116.4(153) | 79.58(118) | 155.9(177.5) | 51.39(63.97) |
| | 3D-UNet w/ SSPA | 30.57(39.29) | 16.69(26.61) | 31.97(48.5) | 73.39(101.12) | 13.27(19.24) | 14.77(24.94) | 21.66(44.6) | 48.01(109.34) | 22.29(74.36) | 112.28(165.02) | 34.77(57.08) |
| 30 scans | 3D-UNet[MICCAI15] | 10.57(19.29) | 10.88(25.58) | 15.5(44.2) | 7.62(6.41) | 8.38(18.39) | 2.91(4.49) | 15.44(44.77) | 32.95(95.03) | 3.73(2.61) | 11.75(45.95) | 13.44(31.07) |
| | 3D-UNet w/ SSPA | 36.16(46.47) | 9.17(22.99) | 15.21(44.6) | 10.53(20.6) | 6.44(6.9) | 3.14(6.59) | 15.96(43.63) | 10.53(20.6) | 3.18(3.05) | 4.76(3.75) | 10.96(24.6) |
| Total | 3D-UNet[MICCAI15] | 0.77(2.18) | 0.26(1.23) | 8.2(44.33) | 3.91(9.8) | 2.49(4.65) | 1.34(0.81) | 8.83(43.97) | 13.73(62.01) | 7.28(44.07) | 2.32(3.23) | 4.83(21.94) |
| | 3D-UNet w/ SSPA | 1.32(5.19) | 0.17(0.77) | 9.33(44.45) | 4.04(10.5) | 2.44(4.03) | 1.38(0.86) | 8.76(44) | 13.18(61.98) | 2.21(1.79) | 2.37(1.91) | 4.35(17.27) |

ACDC and AMOS datasets respectively. Especially our SSPA can bring huge improvements in the segmentation of small organs like GBL, RAG, and LAG, as shown in Tab. 8.

## C.4. Study on the variability of the labeled data.

For semi-supervised learning, the selection of the labeled data can greatly affect the model's performance. Therefore, to further study the influence of different limited labeled data on our method, we randomly selected three subsets of data based on the experimental settings of the ACDC dataset with 3 labeled cases and compared the experimental results with Baseline UNet (Ronneberger et al., 2015), the current SOTA method CTCT (Luo et al., 2022a). Specifically, three selected subsets have a different number of 2D slices, which are 58, 56 and 58, respectively. And the original split of 3 labeled cases contains 68 slices. As shown by the experimental results in Tab. 9, different subset selections influence the model's performance more or less. Compared with the training results from the original split, both UNet and CTCT have larger fluctuations on both DSC and HD95. Although the performance of our proposed method has slightly declined (fewer 2D slices were used), the overall result is more stable and remains superior to the SOTA method. Experimental results further demonstrate the effectiveness of our proposed method.

Table 9: Comparisons of the SOTA method on the ACDC using CNN-Based models under 3 different labeled subsets (not overlapped). (DSC, HD95) from the original split, UNet:(54.34, 47.25), CTCT:(74.16, 18.27), Ours: (82.37, 4.28)

| Labeled | Trans-Based | RV | | Myo | | LV | | Mean | |
|---|---|---|---|---|---|---|---|---|---|
| | | DSC ↑ | HD95 ↓ | DSC ↑ | HD95 ↓ | DSC ↑ | HD95 ↓ | DSC ↑ | HD95 ↓ |
| subset 1 | UNet[MIDL22] | 45.36(30.56) | 97.22(152.77) | 49.42(28.46) | 56.17(124.6) | 58.9(31.9) | 60.13(131.56) | 51.23(30.31) | 71.17(136.32) |
| | CTCT[MIDL22] | 67.45(27.5) | 40.32(110.97) | 70.32(24.78) | 3.81(6.06) | 78.02(27.33) | 21.53(80.79) | 71.93(26.54) | 21.89(65.94) |
| | Ours | 77.09(14.72) | 4.05(3.82) | 77.91(18.91) | 2.69(4.19) | 85.46(18.2) | 3.02(5.69) | 80.15(17.28) | 3.25(4.57) |
| subset 2 | UNet[MIDL22] | 42.86(32.68) | 44.14(55.27) | 55.98(21.43) | 15.5(19.45) | 60.82(25.62) | 33.61(33.55) | 53.22(26.58) | 31.08(36.09) |
| | CTCT[MIDL22] | 69.62(24.8) | 18.48(66.43) | 76.93(11.94) | 4.3(8.34) | 86.47(10.78) | 5.52(9.79) | 77.67(15.84) | 9.43(28.19) |
| | Ours | 75.98(14.69) | 8.21(10.92) | 79.68(8.18) | 4.75(11.39) | 86.8(11.12) | 4.2(7.44) | 80.82(11.33) | 5.72(9.92) |
| subset 3 | UNet[MIDL22] | 47.51(30.06) | 22.54(48.41) | 57.82(25.11) | 7.5(11.86) | 71.57(27.4) | 23.98(80.59) | 58.97(27.53) | 18(46.96) |
| | CTCT[MIDL22] | 69.51(27.47) | 25.03(80.51) | 69.72(23.24) | 4.26(6.04) | 78.39(25.51) | 17.65(66.49) | 72.54(25.41) | 15.65(51.01) |
| | Ours | 75.53(24.31) | 5.46(8.12) | 75.9(17.8) | 3.75(6.07) | 83.26(20.66) | 12.43(48.56) | 78.23(20.93) | 7.21(20.92) |

## C.5. Effects of the number of unlabeled data used.

Normally, the amount of unlabeled data is very large in most datasets. Therefore, most semi-supervised learning methods utilize massive unlabeled data to assist training. However, in scenarios where the amount of unlabeled data is limited, the sensitivity and robustness of the model to the number of unlabeled data are particularly important. Therefore, exploring the impact of the amount of unlabeled data on the model's performance is significant. In this section, we conduct experiments on the ACDC dataset under 3 and 7 labeled cases using a different partition of unlabeled data. Specifically, we use 1/32, 1/16, 1/8, 1/4 and 1/2 partition protocols. When using 1/32 unlabeled data, the number of scans is 2, and 1/16 is 4. Detailed experimental results can be found in Tab. 10.

Surprisingly, our model can achieve the best performance under the 3 labeled case settings with only 1/2 unlabeled data. Also, using only 1/16 and 1/8 of the unlabeled data (8 and 16 volumes) slightly degrades the model. And the performance will drop considerably when using 1/32 of the unlabeled data (DSC from 82.37 to 71.24 and HD95 from 4.28 to 14.7). Except for the 1/32 partition, our model is better than the current SOTA method CTCT under the other five settings.

When trained with 7 labeled cases, the number of unlabeled data has less impact on the performance of our proposed model. When only 1/32 of the unlabeled data is used, our model obtains comparable results to the current SOTA method CTCT, and it is even better on HD95 (from 5.43 to 3.7). In summary, when the amount of labeled data is larger, the impact of the amount of unlabeled data on the model is smaller. Besides, our model only requires a small amount of unlabeled data to assist training and still can achieve satisfactory performance, further illustrating our proposed method's effectiveness.

Table 10: Effects of the number of unlabeled data used on the ACDC dataset. Mean and standard variance (in parentheses) are presented in the table.

| Labeled | Mean | Unlabeled partition used | | | | | | |
|---|---|---|---|---|---|---|---|---|
| | | 1/32 | 1/16 | 1/8 | 1/4 | 1/2 | ALL | ALL(CTCT) |
| 3 cases | DSC ↑ | 71.24(23.91) | 79.47(14.47) | 78.01(18.3) | 76.4(19.69) | **82.63(11.63)** | 82.37(13.79) | 74.16(24.54) |
| | HD95 ↓ | 14.7(35.65) | 8.76(23.06) | 8.8(15.22) | 8.06(21.21) | 4.76(9.45) | **4.28(6.92)** | 18.27(68.85) |
| 7 cases | DSC ↑ | 86.46(8.84) | 86.34(8.95) | 87.48(6.88) | 86.57(8.5) | 85.96(8.81) | **88.98(6.84)** | 86.78(7.78) |
| | HD95 ↓ | 3.7(6.99) | 4.56(8.86) | 4.49(9.08) | 3.5(6.92) | 3.67(6.97) | **2.27(3.43)** | 5.43(10.58) |

Table 11: Comparisons of the SOTA methods on the AMOS using CNN-Based models (Complete results with all 15 categories). Mean and standard variance (in parentheses) are presented in the table.

**DSC ↑**

| Labeled | CNN-Based | SPL | Large | | | | | AOR | Medium | | DUO | PRO/UTE | GBL | ESO | Small | | Mean |
| --- | --- | --- | --- | --- | --- | --- | --- | --- | --- | --- | --- | --- | --- | --- | --- | --- | --- |
| | | | LIV | STO | RKI | LKI | BLA | | PAN | IVC | | | | | RAG | LAG | |
| 15 scans | 3D-UNet[MICCAI19] | 14.98(13.84) | 73.37(11.66) | 19.02(17.17) | 45.4(24.02) | 77.43(14.4) | 15.38(16.27) | 73.08(13.47) | 20.94(17.44) | 58.01(16.92) | 14.69(12.83) | 24.04(27.16) | 7.44(17.9) | 53.59(20.54) | 15.98(15.32) | 13.36(20.14) | 31.34(17.6) |
| | Ent-Mini[CVPR19] | 76.26(17.08) | 80.94(9.41) | 48.64(19.49) | 73.42(19.24) | 66.96(21.08) | 42.08(21.34) | 80.94(11.47) | 30.61(16.9) | 59.25(16.2) | 15.84(14.44) | **41.97(28.45)** | 23.32(22.74) | 51.77(19.8) | 29.78(20.19) | 24.58(23.33) | 49.76(18.74) |
| | CTCT[MIDL22] | 73.48(12.15) | 70.13(10.07) | 39.16(17.76) | **73.45(19.11)** | 69.15(14.66) | 27.79(20.76) | 71.37(12.88) | 22.49(14.31) | 50.92(12.4) | 13.72(12.91) | 9.96(18.8) | 4.29(20.25) | 50.35(20.27) | 13.04(14.89) | 23.04(19.99) | 40.82(16.08) |
| | **Ours** | **88.11(12.38)** | **89.45(6.92)** | **60.29(22.28)** | 70.77(24.3) | **79.87(19.35)** | **51.41(27.71)** | **87.15(9.81)** | **62.47(16.86)** | **69.98(11.99)** | **42.54(19.1)** | 34.87(26.5) | **40.26(29.7)** | **59.46(19.15)** | **44.7(21.21)** | **40.21(25.21)** | **61.44(19.5)** |
| 30 scans | 3D-UNet[MICCAI19] | 71.83(21.51) | 90.94(8.88) | 68.14(23.86) | 84.36(13.18) | 70.61(19.08) | 62.81(25.46) | 88.59(9.82) | 67.99(16.48) | 81.95(7.96) | 50.74(20.28) | 43.79(25.84) | **65.05(30.36)** | 67.09(17.73) | 57.11(17.37) | 56.03(21.07) | 68.47(18.59) |
| | Ent-Mini[CVPR19] | 83.21(15.77) | 90.07(8.44) | 62.92(25.31) | 81.66(18.01) | 81.52(19.11) | 52.02(28.86) | 76.28(16.96) | 41.45(22.45) | 72.78(11.7) | 42(20.39) | 39.84(28.97) | 41.64(30.19) | 53.14(21.69) | 47.83(22.61) | 29.79(23.21) | 59.74(20.91) |
| | CTCT[MIDL22] | **89.39(7.67)** | **93.32(5.51)** | 65.95(22.58) | **90.59(12.68)** | 89.99(6.98) | 55.21(23.32) | 88.31(9.47) | 65.58(14.33) | 77.12(8.4) | 52.03(15.96) | 52.28(24.33) | 63.38(25.03) | 61.72(21.75) | **64.74(13.96)** | 46.52(22.6) | 70.41(15.64) |
| | **Ours** | 81.63(15.2) | 92.24(7.3) | **75.01(20.85)** | 89.57(13.18) | **91.22(10.12)** | **67.81(27.34)** | **90.77(7.33)** | **70.34(15.67)** | **83.78(6.84)** | **58.56(19.77)** | **66.78(21.84)** | 64.5(29.43) | **70.4(17.08)** | 61.15(16.31) | **58.44(20.12)** | **74.81(16.56)** |
| Total | 3D-UNet[MICCAI19] | 94.21(5.94) | 96.43(2.27) | 87.89(15.81) | 94.14(11.43) | 95.47(2) | 83.53(18.9) | 93.59(5.28) | 83.27(11.51) | 88.02(4.78) | 77.43(14.4) | 78.69(19.45) | 80.61(23.38) | 78.99(13.72) | 71.83(13.35) | 73.43(13.45) | 85.17(11.71) |

**HD95 ↓**

| Labeled | CNN-Based | SPL | Large | | | | | AOR | Medium | | DUO | PRO/UTE | GBL | ESO | Small | | Mean |
| --- | --- | --- | --- | --- | --- | --- | --- | --- | --- | --- | --- | --- | --- | --- | --- | --- | --- |
| | | | LIV | STO | RKI | LKI | BLA | | PAN | IVC | | | | | RAG | LAG | |
| 15 scans | 3D-UNet[MICCAI19] | 39.69(31.09) | 17.99(16.85) | 64.2(62.24) | 31.77(50.39) | 39.09(36.75) | 41.39(62.47) | 48.25(32.61) | 27.14(19.27) | 15.46(27.51) | 41.36(52.49) | 38.33(83.71) | 116.4(153) | 14.3(35.87) | 79.58(118) | 155.9(177.5) | 51.39(63.97) |
| | Ent-Mini[CVPR19] | 13.58(28.33) | 13.89(24.21) | 50.1(72.23) | **12.52(45.61)** | 8.44(14.73) | **17.57(43.32)** | 6.73(11.36) | 21.29(43.16) | 9.26(15.06) | 26.44(43.58) | 34.24(75.49) | 55.83(86.59) | 14.2(26.92) | **18.6(47.28)** | 87.21(149.65) | 25.99(48.5) |
| | CTCT[MIDL22] | 33.98(42.47) | 11.11(11.52) | 37.45(52.03) | 33.98(42.47) | 40.54(48.89) | 25.17(43.8) | 40.72(32.38) | 36.52(31.28) | 73.3(23.62) | 39.49(45.24) | 95.29(140.81) | 357.14(75.57) | 10.12(11.47) | 108.05(161.82) | 35.74(71.12) | 63.98(56.06) |
| | **Ours** | **9.13(23.78)** | **9.33(21.83)** | **22.49(47.53)** | 15.47(48.25) | **6.43(16.48)** | 19.83(47.56) | **3.05(6.99)** | **10.41(21.27)** | **6.44(8.38)** | **16.75(43.54)** | **32.97(74.74)** | **33.91(86.06)** | **5.25(5.63)** | 23.78(74.81) | 52.56(123.2) | **17.85(43.27)** |
| 30 scans | 3D-UNet[MICCAI19] | 10.57(19.29) | 10.88(25.58) | 15.5(44.2) | 32.45(52.93) | 9.49(15.47) | **7.62(6.41)** | 3.41(7.53) | 8.38(8.39) | 2.91(4.49) | 15.44(44.77) | 29.96(74.38) | 32.95(95.03) | **6.61(8.99)** | 3.73(2.61) | 11.75(45.95) | 18.8(43.37) |
| | Ent-Mini[CVPR19] | 16.24(32.13) | 7.87(18.29) | 17.99(44.53) | **4.29(12.13)** | 5.06(12.02) | 16.5(43.88) | 9.27(11.43) | 15.73(17.97) | 3.98(2.8) | 17.96(43.63) | 46.59(102.35) | 31.86(84.70) | 29.41(53.42) | 15.31(61.48) | 44.01(109.77) | 10.34(30.2) |
| | CTCT[MIDL22] | **7.58(21.85)** | **4.61(17.33)** | 15.38(44.18) | 6.98(44.24) | **3.28(11.47)** | 16.6(44.04) | 3.67(9.12) | 7.25(7.04) | 3.23(2.57) | 13.96(43.52) | 27.61(74.14) | **23.45(76.08)** | 7.88(11.61) | **2.81(2.1)** | 10.87(43.75) | |
| | **Ours** | **7.77(15.91)** | 9.76(23.83) | **13.5(44.51)** | 11.89(47.35) | 8.07(21.22) | 8.19(11.06) | **2.11(5.04)** | **5.32(5.44)** | **1.91(1.29)** | **13.46(43.96)** | **20.87(64.23)** | 26.22(85.68) | 6.67(13.37) | 3.41(3.26) | **8.89(43.91)** | **9.87(28.68)** |
| Total | 3D-UNet[MICCAI19] | 0.77(2.18) | 0.26(1.23) | 8.2(44.33) | 5.62(44.25) | 0.27(0.49) | 3.91(9.8) | 0.78(1.54) | 2.49(4.65) | 1.34(0.81) | 8.83(43.97) | 13.86(61.82) | 13.73(62.01) | 2.8(4.67) | 7.28(44.07) | 2.32(3.23) | 4.83(21.94) |

## Appendix D.  Additional visualization results.

Here we visualized more results of the test data from ACDC, BraTS and AMOS to further indicate our proposed method's effectiveness. As shown in Fig. 3, Fig. 4, Fig. 5 and Fig. 6, multiple results have proven the discriminability of our method towards different categories, especially for the hard-to-segment medium, small size organs. Such as the right ventricle (RV) in ACDC datasets and adrenal glands (RAG, LAG), and gallbladder (GBL) in the AMOS dataset.

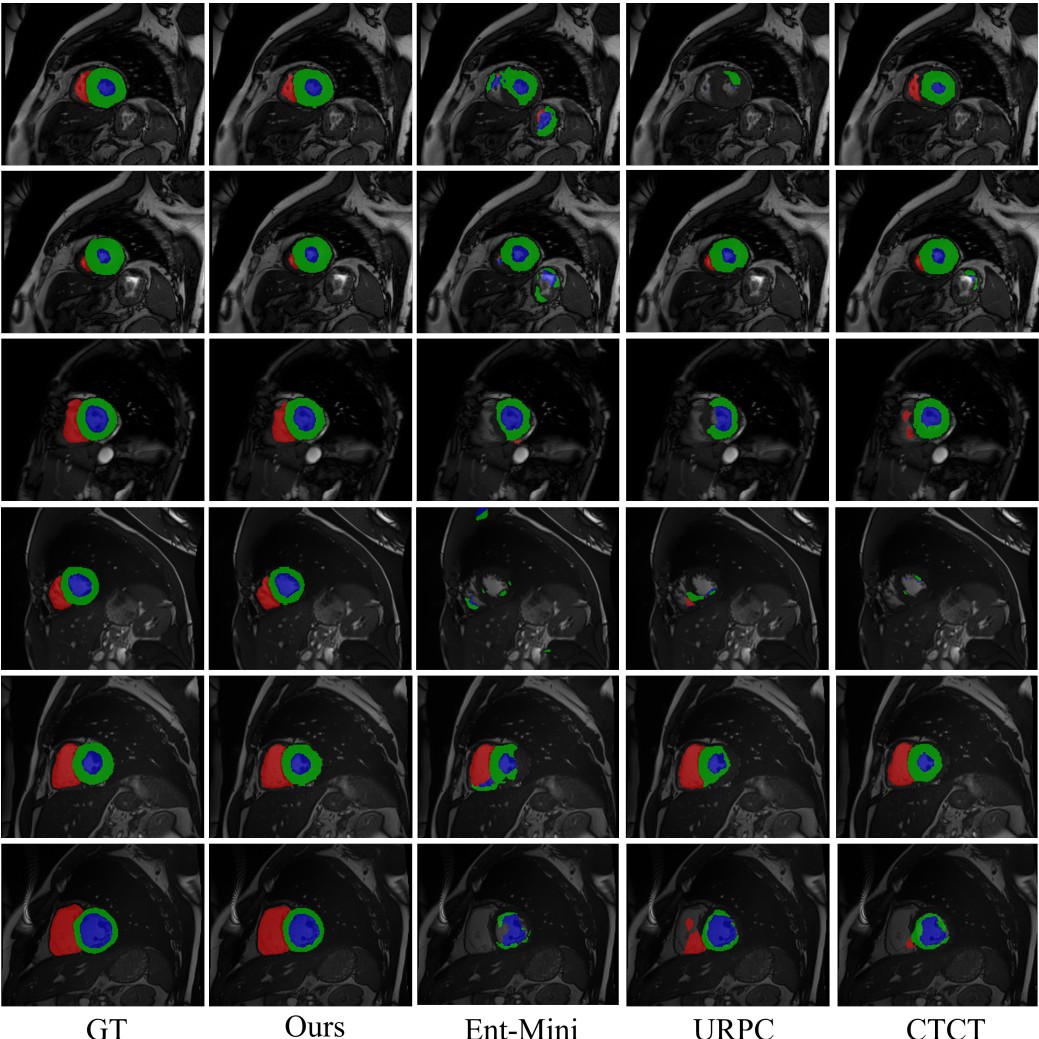

|      GT      |     Ours     |    Ent-Mini    |     URPC     |     CTCT     |

Figure 3: Visualization of ACDC dataset under 3 labeled cases.

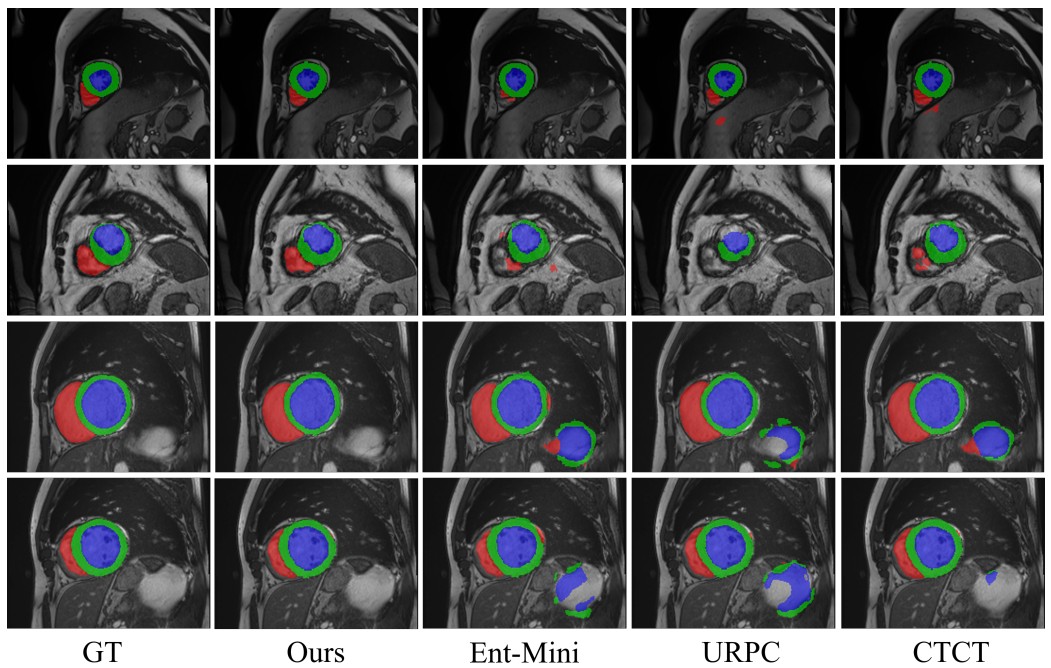

Figure 4: Visualization of ACDC dataset under 7 labeled cases.

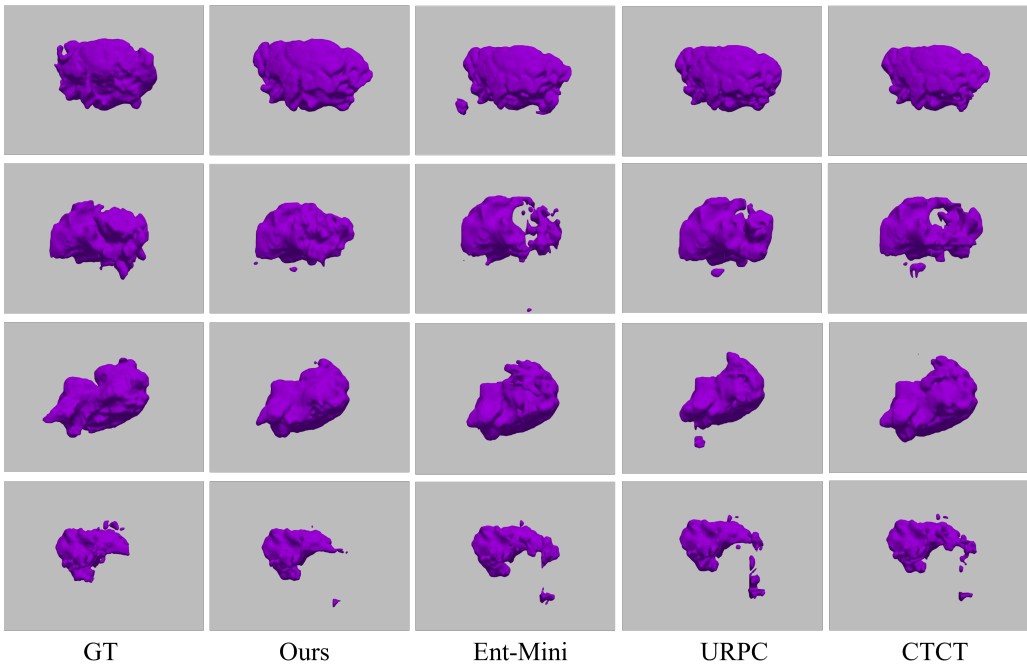

Figure 5: Visualization of BraTS dataset under 25 labeled scans.

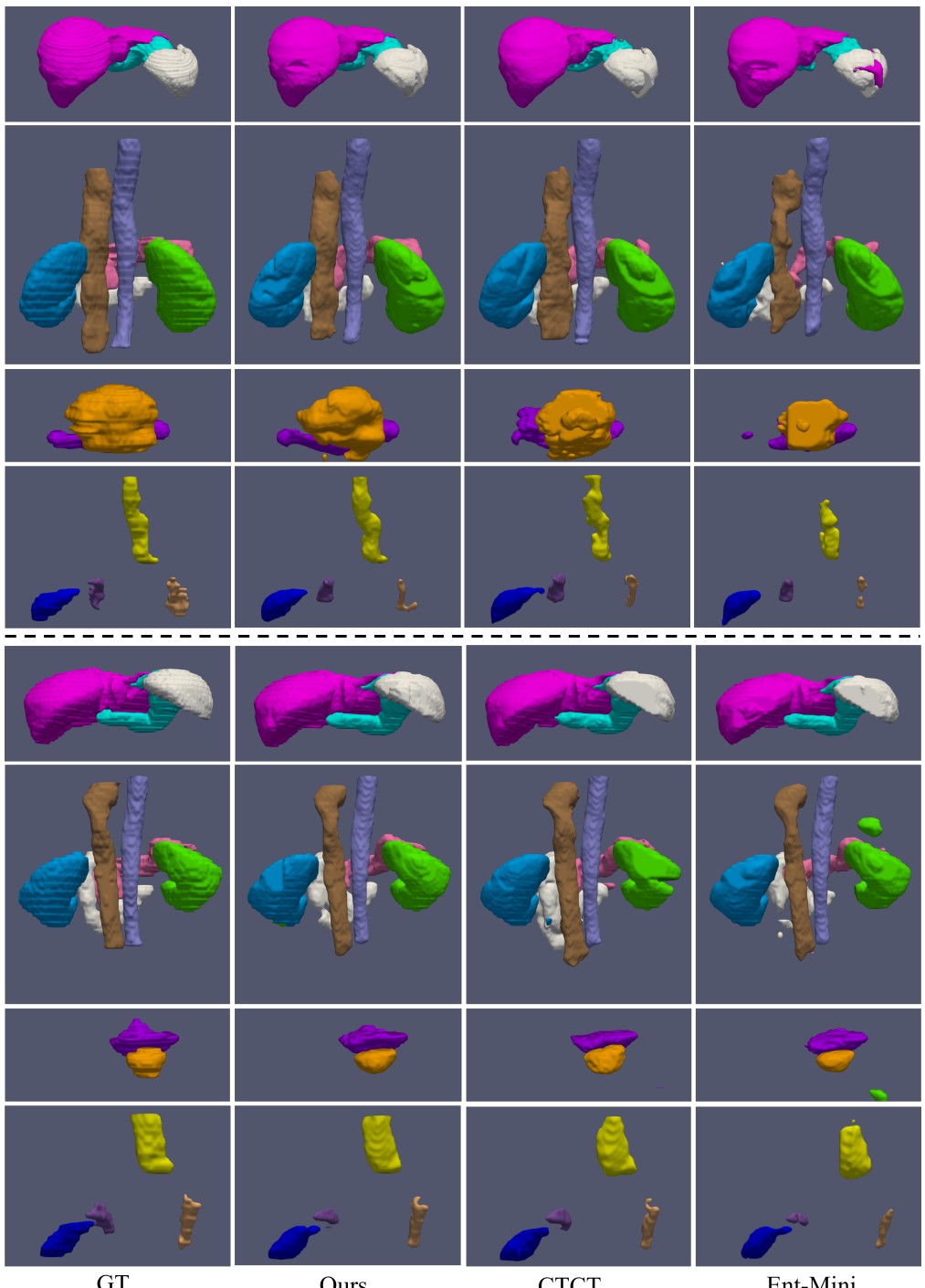

Figure 6: Visualization of AMOS dataset under 30 labeled scans.

