# OpenReview forum: "Inherent Consistent Learning for Accurate Semi-supervised Medical Image Segmentation"
_MIDL.io/2023/Conference — MIDL 2023 Oral_

### Official Review · Reviewer_T5wR · 2023-02-01

**Confidence:** 3
**Preliminary Rating:** 4
**Recommendation:** Oral, Poster

**Summary:**

This paper explores learning in low data settings. The proposed method [Inherent Consistent Learning (ICL)] is evaluated on two public datasets [AMIOS, ACDC] against a robust set of state of the art alternatives. The methods show substantive improvement when learning on extremely limited datasets. Interestingly, benefits decrease relative to standard baselines (Unit) as the number of scans increases to the full number available. Overall, the proposed innovations are highly relevant for low-shot learning.

Please note that no material beyond the references was reviewed per MIDL policy.

**Strengths:**

+ Very strong validation approach across metrics evaluated, structures evaluated, and statistics shown.
+ Clear writing and mathematical presentation.
+ Reproducibly defined loss functions.
+ The approach is quite generalizable toward other

**Weaknesses:**

- The text of the paper is over-optimistic in terms of impact and scope. The scope is good, but many statements are overly definitive. For example: "To solve the above issue..." and "Extensive experimental results...".

- The novelty of the component networks is good, but not transformative.

**Deanonymize Review:**

no

**Paper Type:**

methodological development

**Questions To Address In The Rebuttal:**

Can the network designs be more clearly compared and contrasted to the networks that inspired the work?

Can the tone of the paper be adjusted to match the scope of the existing problem and the level of contribution from the proposed innovation?

---

### Official Review · Reviewer_cRsu · 2023-02-04

**Confidence:** 3
**Preliminary Rating:** 4
**Recommendation:** Oral

**Summary:**

Label efficient segmentation is an important problem in medical image segmentation. This work presents a new semi-supervised segmentation strategy that uses limited labeled data with large amounts of unlabelled data to learn discriminatory representations. This is achieved using cross-attention between the labeled and unlabelled representations. The queries (semantic proxies) thus learned are then used to improve segmentation performance. Experiments on three datasets (one in Appendix) show large performance improvements compared to other semi-supervised learning frameworks, especially in the low labeled regimes.


**Strengths:**

* The proposed use of cross-attention between labeled and unlabeled representations to improve semantic discrimination is an interesting idea. Coupled with the unsupervised consistency learner, enables learning of useful semantic representations at different scales.

* The experimental evaluation is thorough, and shows clear performance improvements at small label regimes, across all the datasets.

* The qualitative discussions and ablation studies presented in Sec 3.5 are very insightful; that the learnt semantic proxies are so clearly separated is highly interesting.

* The literature review is comprehensive and the paper is clearly written.

**Weaknesses:**

* **Variability of labeled data**: In most of the semi-supervised learning settings, the selection of the labels that perhaps have most useful information can make a difference in the number of labeled examples to be used. While the results across the board are better compared to other baselines, an evaluation of the robustness of these performance improvements over different subsets of labeled data can show the usefulness of the learned semantic proxies.

* **Influence of number of unlabeled data points**: Another interesting insight one could obtain is by studying the influence of the number of unlabeled data points used for the unsupervised training.




**Deanonymize Review:**

no

**Detailed Comments:**

See comments above.

**Paper Type:**

both

**Questions To Address In The Rebuttal:**

For both the points mentioned in weakness, studying the influence of the following can be useful:
* varying the subset of labeled data used
* the number of unlabeled data used

While experimental evaluations can be interesting, I can be sufficiently convinced with reasonable speculation.

---

### Official Review · Reviewer_nrmj · 2023-02-06

**Confidence:** 4
**Preliminary Rating:** 5
**Recommendation:** Oral

**Summary:**

The paper presents a novel way of inherent consistent learning for accurate semi-supervised medical image segmentation.
In particular, they introduce two external modules. One supervised semantic proxy adaptor (SSPA) and an unsupervised semantic consistent learner (USCL). Both modules are based on attention mechanism to align the semantic category representations of labeled and unlabeled data as well as updating the global semantic representations over the entire training set.
The evaluation is very extensive comparing it against the best state-of-the-art methods. Moreover, the methods presented in this paper have much better accuracy in most of the experiments, especially when the annotated data is extremely limited.



**Strengths:**

The evaluation is extensively performed which gives confidence on the results.
The quality results are also very nice in comparison to other methods, even in difficult tasks such as automatic segmentation of the RV.
The ability of learning to segment from such a few number of labeled datasets (3 vs 140) while achieving > 82% of dice score while keeping the lowest variance is impressive.
The fact of using a plug-and-play scheme makes the method useful for other architectures.
Moreover the paper is really well-written.
I think it would be very useful for the community to have this paper published.

**Weaknesses:**

A possible weakness might be the non-publication of the code. It is a nice paper and the results are very promising. It would be great if the code would be made available for the community.
Another possible weakness might be the choice of the labelled datasets. It would be great to see whether the results repeat when cross-evaluating with different labelled datasets in order to avoid overfitting. (For example, to see whether 3 other labelled datasets would get the same results on ACDC).

**Deanonymize Review:**

no

**Detailed Comments:**

The caption of Figure 1 doesn't explain much what A and B are even though this is explained in the text.

**Paper Type:**

methodological development

**Questions To Address In The Rebuttal:**

Could you comment on the discussion, how the possible choice of the small labelled datasets could modify the performance of the method?
Could you extend the explanation on the figure captions and make Figure 1 a bit bigger?

---

### Official Review · Reviewer_P4hx · 2023-02-08

**Confidence:** 4
**Preliminary Rating:** 5
**Recommendation:** Oral

**Summary:**

This paper proposes an "Inherent Consistent Learning" (ICL) framework to work on scarce labeled data and numerous unlabeled data They introduce 2  modules based on cross-attention mechanism:  Supervised Semantic Proxy Adaptor (SSPA) and Unsupervised Semantic Consistent Learner.

Tested extensively on 3 large public cohorts.




**Strengths:**

Excellent paper very clearly written with extensive appendix material and method tested extensively on 3 large public cohorts.
Nice visual illustrations for several use-cases.
Clear tables of results.

**Weaknesses:**

Limitations:
- Report training times for the different SOTA and the proposed method
- Unsupervised hyper-parameters have very different values for the 3 use cases. Any insight on their tuning strategy and why this is the case?

**Deanonymize Review:**

no

**Paper Type:**

validation/application paper

**Questions To Address In The Rebuttal:**

Limitations:
- Report training times for the different SOTA and the proposed method
- Unsupervised hyper-parameters have very different values for the 3 use cases. Any insight on their tuning strategy and why this is the case?

---

### Meta-Review · Area_Chair_UcBv · 2023-02-24

**Recommendation:** Accept (Oral)
**Confidence:** 5

**Metareview:**

Due to high cost of medical image annotations, semi-supervised medical image segmentation using limited labeled data is an important problem. The paper presents a novel way of inherent consistent learning for accurate semi-supervised medical image segmentation using  limited labeled data and large amounts of unlabelled data. Two modules based on cross-attention mechanism are introduced: Supervised Semantic Proxy Adaptor (SSPA) and Unsupervised Semantic Consistent Learner. Both modules are based on attention mechanism to align the semantic category representations of labeled and unlabeled data as well as updating the global semantic representations over the entire training set. All reviewers agreed that the paper is clearly written and  evaluations are extensive on 3 large public cohorts. Results indicated improved accuracy especially when the annotated data is extremely limited. All reviewers suggested acceptance of the paper, and I agree with the reviewers.